# Differentiated Effect of Smoking on Disease Activity and Quality of Life among Different Spondyloarthritis Phenotypes

**DOI:** 10.3390/jcm12020551

**Published:** 2023-01-10

**Authors:** Sara Alonso-Castro, Andrea García-Valle, Isla Morante-Bolado, Ignacio Braña, Estefanía Pardo, Rubén Queiro

**Affiliations:** 1Rheumatology Division, Hospital Universitario Central de Asturias, 33011 Oviedo, Spain; 2Rheumatology Division, Complejo Hospitalario de Palencia, 34004 Palencia, Spain; 3Rheumatology Division, Hospital de Sierrallana, 39300 Torrelavega, Spain; 4ISPA Translational Immunology Division, 33011 Oviedo, Spain; 5School of Medicine, Oviedo University, 33011 Oviedo, Spain

**Keywords:** axial spondyloarthritis, psoriatic arthritis, disease activity, quality of life, smoking, cardiovascular risk factors

## Abstract

Background and aims: The effect of smoking on disease activity and quality of life (QoL) in spondyloarthritis (SpA) is far from clear. We aimed to evaluate the relationship between smoking and these outcomes in patients with axial SpA (axSpA) and psoriatic arthritis (PsA). Patients and methods: This cross-sectional observational multicenter study included 242 patients with axSpA and 90 with PsA. The association between conventional cardiovascular risk factors and disease activity as well as QoL, in both SpA phenotypes was evaluated. For this, univariate and multivariate regression analyses were performed, as well as confirmatory meta-analyses. Results: Regardless of age, sex, or disease duration, patients with axSpA showed significantly less association with obesity (OR 0.50 (0.26–0.96), *p* = 0.03) and hypertension (OR 0.33 (0.18–0.62), *p* = 0.0005). However, axSpA was significantly associated with smoking (OR 2.62 (1.36–5.04), *p* = 0.004). Patients with axSpA were more likely to be in a category of high disease activity compared with PsA (OR 2.86, *p* = 0.0006). Regardless of sex, age, disease duration, and education level, smoking was significantly associated with higher disease activity in axSpA (OR 1.88, *p* = 0.027). A fixed-effects model meta-analysis (OR 1.70, *p* = 0.038) confirmed the association between tobacco and disease activity. No relationship was found between smoking (or other cardiometabolic risk factors) and structural damage or worse QoL in either disease. Conclusions: Although the cardiometabolic risk profile is clearly different between both SpA phenotypes, the only clear link between these factors and increased disease activity was observed between smoking and axSpA. Our findings need further confirmation.

## 1. Introduction

The concept of spondyloarthritis (SpA) has been substantiated around entities that share etiopathogenic mechanisms, phenotypic characteristics, and a common genetic basis. This conceptual umbrella encompasses entities where symptoms and signs of involvement of the axial skeleton predominate, such as radiographic (r) and non-radiographic (nr) axial SpA (axSpA), along with others such as psoriatic arthritis (PsA), where the inflammatory manifestations of the peripheral skeleton are more prevalent [1]. In both SpA groups, the physical function and quality of life (QoL) of patients are seriously compromised. These patients also manifest serious impediments to their social participation and work capacity. Consequently, the resulting costs are enormous, both individually and socially [1,2,3].

In recent decades, an enormous effort has been made to design, validate, and put into clinical practice numerous tools to evaluate the activity of these diseases in their different aspects, as well as the impact they generate on the QoL of patients [1,4]. It has also become clear that measures of activity (mostly focused on the doctor’s perspective) do not always match the measures of impact on QoL (focused on the patient’s perspective), so efforts continue to be made to bring both visions closer together [5,6].

Spondyloarthritides are usually accompanied by a series of comorbidities, most of them belonging to the sphere of cardiometabolic health [7]. Cardiometabolic comorbidities, together with certain factors linked to lifestyle (i.e., smoking), not only compromise life expectancy, but can also be linked to greater disease activity, greater structural damage, worse response/persistence of biological therapies, and potentially worse QoL [7,8,9,10,11]. However, an important part of the literature on the associations between cardiometabolic factors and the above aspects is not only contradictory but is surely limited by several biases [11]. For example, some cardiovascular risk factors (e.g., tobacco, obesity) seem to be associated with greater disease activity in axSpA and PsA; however, when the treatment goals are analyzed in patients undergoing conventional and biological systemic therapies, it does not seem that these factors are associated with poorer therapeutic results [11,12]. On the other hand, although tobacco is a clear risk factor for PsA in the general population, psoriatic smoking patients seem to be “protected” against the development of joint manifestations [11]. These and other “observational paradoxes” are in part linked to confounding factors and biases (i.e., collider bias) that are often overlooked in many of these studies [11].

For all the above, it is still necessary to examine more deeply the connections between cardiometabolic risk factors and the activity and the impact on QoL in patients with SpA. In the present work conducted in patients with axSpA and PsA we have analyzed the association between conventional cardiovascular risk factors and disease activity, as well as with health impact, evaluating both disease aspects through validated outcome measures.

## 2. Patients and Methods

This is a post hoc comparative analysis of three multicenter observational cross-sectional studies in which we verified the construct and discriminant validity of the Assessment of SpondyloArthritis international Society-Health Index (ASAS HI) in axSpA [13] and PsA [14], as well as the construct/discriminant validity of the Routine Assessment of Patient Index Data 3 (RAPID3) in axSpA [15]. The inclusion/exclusion criteria, methodological details, main results, and ethical considerations applicable to these studies have been published elsewhere [13,14,15]. This study complies with the declaration of Helsinki. The ethics committees of the Hospital Universitario Central de Asturias (Oviedo, Spain), the Hospital Sierrallana (Torrelavega, Spain), and the Palencia Healthcare Complex (Palencia, Spain) approved the research protocol of each study. Informed consent was obtained from the subjects of these studies.

Briefly, detailed data on the family history of the disease, educational level, comorbidities, received treatments, analytical data, structural damage, disease sub-phenotypes, and several composite outcome measures were collected. The latter were used as external anchors to validate the psychometric properties of the ASAS HI and RAPID3 in the study subpopulations. Disease activity assessment in axSpA was performed through the Bath Ankylosing Spondylitis Disease Activity Index (BASDAI) and the Ankylosing Spondylitis Disease Activity Score (ASDAS), and for PsA disease activity assessment, we used the Disease Activity index for PsA (DAPSA).

To assess disease impact on QoL outcome measures, we used the ASAS HI in one of the axSpA population and in the PsA group. In the latter, we also used the PsA Impact of Disease (PsAID) questionnaire. The ASAS HI is a linear composite measure based on the International Classification of Functioning, Disability and Health (also known as ICF) containing 17 items which address aspects of pain, emotional functions, sleep, sexual functions, mobility, self-care, and community life. Each positive answer is scored 1, and a negative answer is scored 0. The final score is the sum of individual items so that the higher the score, the greater the negative impact due to the disease. An ASAS HI > 5 is consistent with a moderate to bad health status (high impact) [16]. The PsAID questionnaire reflects the impact of PsA from the patients’ perspective. It comprises 12 physical and psychological domains. The final score ranges from 0 (best status) to 10 (worst status) with a cutoff of 4. A PsAID score below 4 is considered a patient-acceptable symptom state [17]. In a second axSpA cohort, we used the RAPID3. This is an index composed of three patient self-reported measures from the rheumatoid arthritis (RA) core data set: physical function on the Health Assessment Questionnaire (HAQ) or its multidimensional version (MDHAQ), along with pain and the patient’s overall disease assessment on two visual analog scales (VAS; range 0–10). In this study, physical function was assessed by the HAQ. The RAPID3 is calculated as the sum of its three domains, each domain reaching a maximum of 10 points, so that the range is from 0 to 30. Severity categories have been defined for RA as follows: ≤3 for remission, 3.1–6.0 for low, 6.1–12.0 for moderate, and >12 for high severity [18].

The different cardiovascular risk factors analyzed in this study (smoking, obesity, diabetes, hypertension, and dyslipidemia) were categorized as present or absent.

### Statistical Analysis

A descriptive statistical analysis of all variables was performed, including central tendency and dispersion measures for continuous variables, and absolute and relative frequencies for categorical variables. To test the differences between quantitative variables, parametric (Student’s T-test) and non-parametric tests (Mann–Whitney U or Kruskal–Wallis H) were used according to the goodness-of-fit test. Differences between qualitative variables were measured by a Pearson´s chi-square and Fisher’s exact tests. We conducted univariate and multivariate regression models to test for potential associations between each cardiovascular risk factor and the highest disease activity categories (DAPSA > 14, BASDAI > 4, ASDAS > 2.1), as well as with the highest disease impact categories (PsAID ≥ 4, ASAS HI > 5, RAPID3 > 6). All associations were corrected according to age, sex, and disease duration. From the results obtained in the univariate logistic regression models of the different significant associations between cardiovascular risk factors and the above disease outcomes, a summary effect of the cardiometabolic factor on the disease outcome was estimated through meta-analyses assuming a fixed effects model. Statistical significance was set at a value below 5%. Data were analyzed using the R statistical software package.

## 3. Results

### 3.1. Summary of Study Population

Our analysis included 242 patients from two different axSpA cohorts and 90 with PsA. The combined axSpA cohort included 205 men and 37 women of mean age 48.3 ± 12.7 years, and a median disease duration of 9 years (IQR: 4–17). We also included 52 men and 38 women with PsA of mean age 55.3 ± 14.3 years, with a median arthritis duration of 7 years (IQR: 3–14) and a median psoriasis duration of 16 years (IQR: 10–29). The main disease features of these populations have been published elsewhere [13,14,15].

Compared with the axSpA, the patients with PsA were older (55.3 ± 14.3 years versus 48.3 ± 12.7 years, *p* < 0.001) and were receiving fewer biological therapies at the inclusion visit (44.4% versus 63%, *p* < 0.01); however, more PsA subjects were in remission/low activity upon study entry (73% versus 47.6%, *p* < 0.001). The main disease outcomes are presented in Table 1.

#### Distribution of Cardiometabolic Factors between Both SpA Populations

Table 2 shows the frequencies of the traditional cardiometabolic risk factors among the two SpA phenotypes as well as in the whole group.

When comparing both groups, regardless of age, sex, or disease duration, the patients with axSpA showed significantly less association with obesity (OR 0.50 (0.26–0.96), *p* = 0.03) and hypertension (OR 0.33 (0.18–0.62), *p* = 0.0005). However, axSpA was significantly associated with smoking (OR 2.62 (1.36–5.04), *p* = 0.004). The higher prevalence of diabetes among patients with PsA was explained by age (OR 1.07 (1.03–1.10). *p* < 0.0001). We found no differences regarding dyslipidemia. More PsA patients had ≥3 cardiometabolic factors compared with axSpA patients (8% vs. 3%).

### 3.2. Cardiometabolic Factors and Disease Outcomes

In the univariate logistic regression analysis, only smoking was found to be associated with higher disease activity (OR 2.43 (1.30–4.55), *p* = 0.005). When analyzing both SpA populations, the axSpA patients were more likely to be in a category of high disease activity compared with PsA patients (OR 2.86, (1.57–5.20), *p* = 0.0006). Furthermore, in both groups of patients, women were more likely to have highly active disease (axSpA OR 2.29 (1.33–3.96), *p* = 0.003; PsA OR 4.19 (1.55–11.34), *p* = 0.005). An inverse association was found between age and smoking in PsA (OR 0.95 (0.90–0.99), *p* = 0.05). Regardless of sex, age, disease duration, and education level, smoking was significantly associated with more active disease, but only in patients with axSpA (OR 1.88 (1.07–3.28), *p* = 0.027). We then analyzed the interaction between biological therapy and smoking to address whether the effect of smoking on increased disease activity was influenced by these therapies. However, the interaction term was not statistically significant (OR 1.04 (0.29–3.70), *p* = 0.95).

### 3.3. Meta-Analysis of the Tobacco Effect on Disease Activity

To verify that smoking was indeed associated with greater disease activity, we carried out a meta-analysis (grouping the two different axSpA cohorts with the PsA group) based on the results obtained in the univariate logistic regression models assuming a fixed effects model (Figure 1).

According to this analysis, the effect of tobacco on greater disease activity was maintained (OR 1.70 (1.03–2.82), *p* = 0.038). No heterogeneity was detected between cohorts (I^2^ = 0%). No effect was detected between smoking and the structural damage evaluated by the mSASSS score (OR 0.19 (−6.99–7.36), *p* = 0.96). There was also no association between smoking and the presence of bone erosions.

### 3.4. Impact of Cardiometabolic Risk Factors on Quality of Life

None of the cardiovascular risk factors analyzed, nor their number, were associated with higher disease impact as assessed either by PsAID, ASAS HI or RAPID3. With respect to disease impact, smoking was not related to a higher disease impact by any of the instruments listed above. We analyzed the interaction between disease type and smoking to see whether the effect of smoking on a high disease impact was different between those with PsA and axSpA. However, the interaction term was not statistically significant (OR 1.65 (0.40–6.80), *p* = 0.49).

## 4. Discussion

In this secondary analysis, we detected an association between tobacco and axSpA that remained after correcting for age, sex, disease duration, and educational level. We also found a higher prevalence of obesity, hypertension, and diabetes among PsA patients. However, the association with diabetes was due to an older age among PsA patients. We also found an association between smoking and increased disease activity among axSpA patients that remained significant after a meta-analysis of all cohorts. However, we did not find any association between tobacco and structural damage. Neither smoking, nor any other cardiometabolic risk factor, was associated with a higher disease impact, whether it was assessed by ASAS HI, PsAID or RAPID3.

Smoking has been linked to an increased risk of disease, greater disease activity, poorer response/persistence of biological therapies, and even greater structural damage in SpA [11]. Despite this, many factors related to smoking may also influence both the incidence and severity of axSpA. In particular, smoking tends to be associated with manual occupations, unemployment, lower physical activity, lower educational attainment, higher BMI, and other lifestyle and socioeconomic factors [11,19,20]. For example, it has been argued that smoking is related to structural damage progression in axSpA, but also patients with physically demanding jobs tend to have radiographic damage progression despite a smoking habit [11,20]. These findings suggest that occupation could be a more important causal candidate for radiographic progression than smoking. Consistent with this, we did not detect any association between smoking and structural damage assessed by the mSASSS score. In addition, we found that the effect of smoking on disease activity was independent of the educational level, thereby rendering smoking as a true promoter of higher disease activity in our context. Nonetheless, untangling the independent impact of smoking from the above confounding factors remains a major challenge.

On the other hand, tobacco–PsA relationships are complex and even contradictory [11]. Thus, despite a clear association between tobacco and higher risk of PsA in the general population, this effect seems to be lost, and even reversed, when the population with skin psoriasis is considered alone [11]. Therefore, an apparent “protective” effect of tobacco against the development of arthritis has been detected in patients with psoriasis [11]. These contradictions and apparent paradoxes also carry over to studies that have analyzed the connection between smoking and disease activity or structural damage in these conditions [11].

In our PsA cohort, women presented higher levels of disease activity, and an inverse association between smoking and age was also apparent. It has been observed that younger patients with PsA tend to smoke more than their older counterparts. Likewise, the impact of the disease in younger patients also tends to be greater (assessed by the PsAID) in relation to older subjects [21]. Therefore, it has been argued that there is an apparent connection between tobacco and worse functional and quality of life outcomes in this population [21]. Nonetheless, in our study, we have not been able to verify this. In our PsA group, after correcting for age, smoking, and disease duration, the female gender remained associated with a higher level of disease activity, confirming previous results [22,23]. On the other hand, we did not find any association between tobacco and either PsA disease activity or its severity.

In the meta-analysis of axSpA cross-sectional studies, ever smokers had significantly higher BASDAI, BASFI and spinal pain, and poorer QoL than never smokers [11]. In a large Swedish survey of respondents with PsA, ever smokers reported poorer QoL, global health, pain, and fatigue than never smokers [11]. Therefore, there is consistent evidence that SpA severity and tobacco are associated. Our findings are partly consistent with these observations. However, a direction of causality cannot be established through cross-sectional studies such as ours. On the other hand, in axSpA longitudinal studies, although smokers had more severe disease at TNF inhibitor initiation than non-smokers, they experience the same absolute reduction in disease outcomes over time [11]. In line with this, a recent PsA study showed that cardiometabolic factors did not influence the chances of achieving stringent treatment goals [24]. Moreover, a recent multicenter study carried out in patients with axSpA and PsA found better persistence rates of secukinumab in obese, hypertensive, and diabetic patients [25]. However, we do know if interventions aimed at reducing weight are associated with better outcomes in PsA [26]. In addition, in our analysis of tobacco-biological therapy interaction, we did not find any effect of smoking on disease activity or its impact on QoL in either disease.

Smoking remains a major health problem. As such, any rheumatic patient with this habit should be encouraged to quit early. However, the literature that associates smoking with higher disease activity or a higher risk of adverse outcomes in patients with PsA or axSpA is full of counterintuitive and paradoxical findings, surely related to inappropriate study designs, and multiple factors creating both bias and confusion [11].

Our study is a secondary analysis of three cohorts (two with axSpA and one with PsA) initially designed with a different purpose than the mere analysis of the effect of tobacco and other cardiovascular risk factors on several outcomes. However, we included sufficient patients (>300) to perform a meta-analysis of the effect of tobacco on disease activity. After this, we were able to consolidate the initial results of the univariate regression analyses. In any case, we must bear in mind that most of our patients had adequate control of their disease, so few of them were in the highest categories of both activity and negative impact. Furthermore, as we have already commented, the cross-sectional nature of this study makes it impossible to analyze the direction of the associations we found. A major drawback of this study is that smoking was analyzed only as a categorical variable, making it impossible to explore a possible dose-dependent effect of the smoking-activity relationship [11,27]. In fact, in our study, smoking refers to current smokers regardless of the number of cigarettes per day. For this reason, for example, we do not know what role past smoking may have on greater activity (or not) of the disease. Besides its effect on disease activity, the interaction analyzes showed that there was no association between tobacco, or other cardiometabolic risk factors, and a higher impact on QoL. As our results are somewhat contradictory to others previously published, it is necessary to carry out studies with larger samples sizes and of suitable design.

Although we have only been able to corroborate what has already been provided in other studies, the most innovative aspect of our report has been to compare two very different populations of patients with SpA. Thus, we saw how smoking does affect disease activity in axSpA, whereas its role in PsA remains less clear, so more information is still needed on this last point.

In summary, our study confirms a relationship between tobacco and SpA disease activity, but mostly in patients with axSpA. We did not find any association with other adverse disease outcomes, but it is also true that many of our patients had good control of their disease upon study entry. Our results need further confirmation.

## Figures and Tables

**Figure 1 jcm-12-00551-f001:**
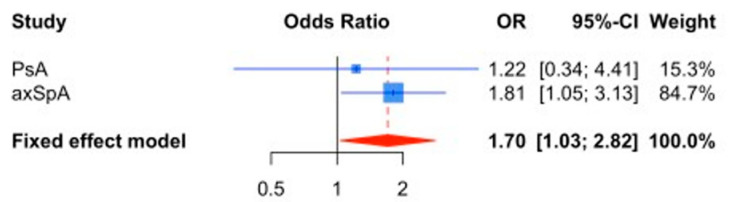
Forest plot of the meta-analysis based on a fixed-effect model performed by pooling the effect of smoking on disease activity.

**Table 1 jcm-12-00551-t001:** Main disease outcomes of both study populations.

Axial Spondyloarthritis	Psoriatic Arthritis
BASDAI 3.59 ± 2.40	DAPSA 9.7 ± 7.8
ASDAS-CRP 2.07 ± 0.83	PsAID 2.8 ± 2.3
ASAS-HI 5.4 ± 3.8	ASAS HI 5.8 ± 4.3
RAPID3 9.45 ± 6.7	ASAS HI-PsAID correlation, *r*: 0.75, 95%CI: 0.64–0-83, *p* < 0.001.

BASDAI: Bath Ankylosing Spondylitis Disease Activity Index; ASDAS-CRP: Ankylosing Spondylitis Disease Activity Score-C-reactive protein; ASAS-HI: Assessment of SpondyloArthritis international Society-Health Index; RAPID3: Routine Assessment of Patient Index Data 3; DAPSA: Disease Activity index for PSoriatic Arthritis; PsAID: Psoriatic Arthritis Impact of Disease.

**Table 2 jcm-12-00551-t002:** Distribution of cardiometabolic risk factors among the study populations.

CM Factor	PsA, n: 90	axSpA, n: 242	Total, n: 332
Obesity, *n* (%) *	21 (23)	28 (12)	49 (15)
Hypertension, *n* (%) *	38 (42)	38 (16)	76 (23)
Diabetes, *n* (%) *	14 (15)	13 (5)	27 (8)
Dyslipidemia, *n* (%)	31 (34)	56 (23)	87 (26)
Smoking, *n* (%) *	13 (14)	81 (33)	94 (28)

CM: cardiometabolic. PsA: psoriatic arthritis. axSpA: axial spondyloarthritis. * Significant statistical differences. See text for detailed explanation.

## Data Availability

The data on which this study is based are stored in databases. This information is available to third parties on reasonable demand.

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
