# Peer review of "Differentiated Effect of Smoking on Disease Activity and Quality of Life among Different Spondyloarthritis Phenotypes"

_jcm, 2023, doi:10.3390/jcm12020551_

Round 1

Reviewer 1 Report

The manuscript by Castro et al. investigated the effect of smoking on disease activity and QoL in axial spondyloarthritis (axSpA) and psoriatic arthritis patients and showed a correlation between smoking and SpA (mostly axSpA) confirming previous published data.

I have the following suggestions on the manuscript:

1)    The data is hard to follow in the present format in the results section. It would be better to present the statistical data of various measured factors in table format comparing the disease outcomes in axSpA and PsA.

2)    The figure in the manuscript (Figure 1) is of low quality and resolution. Please use a high-quality image and expand its size for better clarity.

3)    The major drawback of the manuscript is that no novel data was obtained in the study, and the manuscript only confirms previous findings.

Author Response

  • The data is hard to follow in the present format in the results section. It would be better to present the statistical data of various measured factors in table format comparing the disease outcomes in axSpA and PsA.

R/ Thank you very much for this suggestion. We follow it and incorporate the changes in the new version.

  • The figure in the manuscript (Figure 1) is of low quality and resolution. Please use a high-quality image and expand its size for better clarity.

R/ Ok, thanks. If the manuscript is finally accepted, we will contact the production team to provide better quality figure.

  • The major drawback of the manuscript is that no novel data was obtained in the study, and the manuscript only confirms previous findings.

R/ We agree. Indeed, we have only been able to corroborate what has already been provided in other studies, although the most innovative aspect of our report has been to compare two very different populations of patients with SpA. We thus saw how smoking does affect activity in axial SpA, while its role in PsA is less clear, and therefore more information is needed about it. We added a comment in the discussion.

Reviewer 2 Report

While influence of smoking on many diseases can be described as just harmful, the Authors decided to study its effects in spondyloarthropathies. In my opinion study is interesting and valuable to hospitalists, who should be trying to convince the patient with SpA, that smoking cessation will influence their disease control. Nevertheless one of the drawbacks of the study is using just categorical variable to present the smoking status of the patient instead of more accurate measures which greatly diminishes study quality, so the paragraph about study limitations should be expanded.

Author Response

While influence of smoking on many diseases can be described as just harmful, the Authors decided to study its effects in spondyloarthropathies. In my opinion study is interesting and valuable to hospitalists, who should be trying to convince the patient with SpA, that smoking cessation will influence their disease control. Nevertheless, one of the drawbacks of the study is using just categorical variable to present the smoking status of the patient instead of more accurate measures which greatly diminishes study quality, so the paragraph about study limitations should be expanded.

R/ We completely agree with the reviewer. Indeed, introducing smoking only as a categorical variable makes it impossible to know if there is a dose-dependent effect on the tobacco-activity relationship. Unfortunately, as this was a secondary study, it has not been possible to delve further into this aspect. The comment is extended in the part about weaknesses of the study.